# “They Reckon They’re Man’s Best Friend and I Believe That.” Understanding Relationships with Dogs in Australian Aboriginal Communities to Inform Effective Dog Population Management

**DOI:** 10.3390/ani10050810

**Published:** 2020-05-07

**Authors:** Gemma C Ma, Jason Ford, Lillian Lucas, Jacqueline M Norris, Jessica Spencer, Ann-Margret Withers, Michael P Ward

**Affiliations:** 1Sydney School of Veterinary Science, Faculty of Science, the University of Sydney, Camperdown 2006, Australia; jacqui.norris@sydney.edu.au (J.M.N.); michael.ward@sydney.edu.au (M.P.W.); 2RSPCA NSW, Yagoona 2199, Australia; amwithers@rspcansw.org.au; 3Representative of the Ngemba Nation, Cobar 2835, Australia; jford3757@gmail.com; 4Regional Enterprise Development Institute Ltd. (REDI.E), Dubbo 2830, Australia; lillian.lucas101@gmail.com; 5NSW Health, Aboriginal Environmental Health Unit, Dubbo 2830, Australia; Jessica.Spencer@health.nsw.gov.au

**Keywords:** dog, companion animal, Aboriginal, dog population management

## Abstract

**Simple Summary:**

Dogs are an important part of many communities; however, they can also cause problems, especially when they are able to roam freely. The best way to manage dog populations to avoid health and safety risks to people varies between different societies and communities. We aimed to better understand the role of dogs in Aboriginal communities in Australia, to inform dog health interventions. Interviews with dog owners in nine Aboriginal communities in New South Wales, Australia showed that dogs are valued as companions and guardians and are considered part of the family. Many dogs spend a lot of time with their families, are allowed indoors and often share beds with people. However, many dog owners had trouble accessing veterinary services for their dogs; cost and lack of transport were the most important barriers. Our findings demonstrate the positive role of dogs in Aboriginal families and communities but also highlight major challenges in accessing the veterinary services necessary to effectively maintain dog health and welfare. We also show that different ideas about the role of dogs in different communities should inform how dog management interventions are designed and delivered.

**Abstract:**

Dogs are important companions to people in many societies; however, dogs can also be associated with risks to public health and safety. Dog population management is therefore an important consideration globally. This study aimed to better understand the role of dogs in Aboriginal communities and the barriers to accessing veterinary services. Semi-structured interviews were conducted with 85 dog owners from nine Aboriginal communities across New South Wales, Australia. Many positive aspects of dog ownership were identified and few negatives. Dogs are considered an important part of family and community life and many dogs are allowed indoors (63.4%), even sharing beds with their owners. Most dogs were kept for companionship (84.7%) and/or as guard dogs (45.9%) and all respondents considered their dog part of their family. However, respondents had low levels of engagement with mainstream veterinary services, and many respondents identified significant barriers to accessing veterinary services, especially cost and transport. This study demonstrates the important and positive role of dogs in Aboriginal families and communities but also highlights a significant veterinary service gap. Our findings demonstrate that different perspectives on the role of dogs necessitates a different, culturally inclusive approach to dog management interventions.

## 1. Introduction

Dogs have been companions to humans across the globe for over 33,000 years [1] and living with dogs has been associated with physical, emotional and social benefits [2,3,4]. Hence, managing dog population size and dog health and minimising risks presented by dogs to public health and safety is an important consideration globally. The role of dogs, people’s relationships with dogs, and attitudes towards them vary between communities and cultures [5]. Understanding these differences is essential to facilitate effective management of dog populations to maintain community safety and ensure adequate dog health and welfare. 

Dog ownership is common in Australia, with 39% of households owning at least one dog, the most favoured species of companion animal nationally [6]. Dogs are especially prominent in Australian Aboriginal communities, in which 65% of households claim at least one dog, much higher than the national average [7]. Canids have had an important relationship with Aboriginal people in Australia since arrival of the dingo during the Holocene, some 5000 years ago [8]. The presence of dingoes in Aboriginal camps at the beginning of colonisation was almost ubiquitous [9]. However, dingoes were never fully domesticated. Rather, they were acquired from dens as pups, kept as companions while young and encouraged to return to the wild on reaching sexual maturity [9]. Dingoes hold deep cultural and spiritual significance which permeates every aspect of Aboriginal society [10]. Domestic dogs were readily adopted into Aboriginal communities on the arrival of European colonisers and continue to be held in high regard in many Aboriginal communities, fulfilling a similar cultural role to the dingo [10,11,12]. In some communities, dogs are given skin names (skin names denote a person’s position in the kinship system and establish their relationship to others; they are generally reserved for people); in others, dogs are taken as totems [11] or valued as spiritual protectors [12]. The status and cultural importance of dogs, however, varies within and between different communities and attitudes towards dogs and dog ownership are not static [10,13].

Effective management of domestic dog populations is an important component of maintaining community liveability and health and is identified as a priority by many Aboriginal communities in Australia [13]. Large populations of dogs that roam freely can create a public nuisance and compromise community safety due to dog fights, excessive barking, faecal pollution and begging for and stealing food [11]. The incidence of dog attacks on people and dog bite injuries can be high [14] and transmission of zoonotic infections and parasites including Giardia, scabies, roundworms, hookworms and tapeworms can also occur [15,16]. When poorly managed, dog populations can experience compromised welfare and a substantial burden of preventable disease and suffering, including a high incidence of road traffic accidents and other injuries, increased competition for food, dehydration and infectious disease [17]. However, dog population management faces major challenges when veterinary services are unavailable, unpalatable or unaffordable, and these challenges are exacerbated by remoteness and material disadvantage—circumstances disproportionately experienced by Aboriginal people [18,19].

Dog population management ideally entails humane control of dog population size and growth, management of free-roaming dogs and provision of veterinary services to ensure adequate animal welfare [20]. Effective dog population management relies on a thorough understanding of dog population dynamics in the target population. This necessitates an understanding of how people behave towards dogs and what drives acquisition of dogs. It is also important to establish patterns of confinement, roaming and abandonment of dogs and to investigate how dogs on the street are tolerated and cared for [20]. An effective understanding of dog population dynamics also requires an understanding of the social, cultural and regulatory barriers to the delivery and uptake of services [14]. 

In response to an Aboriginal community-identified need for a dog health intervention, the Royal Society for the Prevention of Cruelty to Animals (RSPCA) New South Wales (NSW) partnered with NSW Health Aboriginal Environmental Health Unit to commence delivery of Indigenous community companion animal health programs (ICCAHPs). ICCAHPs aim to create sustainable improvements in animal welfare and dog population management, particularly in remote areas without access to local veterinary services. However, research to inform the design and implementation of such interventions is limited. This study aimed to better understand patterns of dog ownership and people’s relationships with dogs in Aboriginal communities in New South Wales. The study also aimed to understand key contributors to dog population dynamics and the barriers that exist to accessing veterinary services to inform the design and delivery of best-practice dog population management interventions.

**NB The use of terminology in this manuscript is contested, and authors have different views. ‘First Nations’ is preferred over ‘Aboriginal’ by some of the authors.** In addition, ‘Aboriginal’ is used in place of ‘Aboriginal and Torres Strait Islander’ in recognition that Aboriginal people were the original inhabitants of NSW, where this research took place. No disrespect is intended to Torres Strait Islander people or communities.

## 2. Materials and Methods

### 2.1. Study Context

The study was conducted in nine communities across NSW, Australia that participated in RSPCA ICCAHPs. The ICCAHP involved visits to communities by a multi-disciplinary team consisting of veterinarians, veterinary nurses, community outreach officers and education officers. Eligible participants (recipients of government welfare) were able to access free basic veterinary services including health checks, surgical sterilisation, vaccination, microchipping and parasite treatments. Communities were prioritised to receive an ICCAHP based on the following factors: (1) Aboriginal community-identified need for a dog health intervention, (2) large proportion of the population identifying as Aboriginal and, (3) relative socioeconomic disadvantage based on the Australian Bureau of Statistics (ABS) Index of Relative Socio-Economic Advantage and Disadvantage [21]. According to the ABS Remoteness Structure, the communities ranged from inner-regional (Community 1) and outer-regional (Community 2 & 3) to remote (Community 4 & 5) and very remote (Community 6, 7, 8 & 9) [22]. Local access to veterinary services is often limited. Three of the communities have veterinary clinics present locally, the other communities are between 25 km and more than 200 km from the nearest veterinary clinic (average 111 km). Eligibility for participation in the ICCAHP requires pet owners to be eligible to receive government benefits and to live within the geographical boundary of the participating town or community.

### 2.2. Questionnaire

All dog owners who participated in the ICCAHP were eligible for the study. Participation was optional and was with informed consent. Respondents completed a questionnaire (Appendix A), which was delivered as a semi-structured interview. The questionnaire was developed with community representatives from participating Aboriginal communities to ensure questions reflected community priorities and were appropriately worded. The questionnaire was piloted in two Aboriginal communities while in development prior to the start of the study. Questionnaires were delivered by two of the research team (GM and JS) who were affiliated with the ICCAHP. Dog owners were invited to participate in the study after their pets had received veterinary care through the ICCAHP to avoid a perception that their dog’s care might be affected by their responses. A participatory action approach was used [23], with questions adjusted in response to preliminary findings and experience delivering early questionnaires. The questionnaire consisted of three sections: (1) your relationship with your dog; (2) dogs in the community; and (3) dogs and your house. Dog owner responses to questions and additional comments were recorded verbatim by writing, either on a tablet device or on paper.

### 2.3. Quantitative Data Analysis

Statistical analyses were conducted using SPSS^®^ Statistics Version 24 (IBM^®^, Chicago, IL, USA). Measures of association were calculated using Pearson’s chi-squared tests and odds ratios and 95% confidence intervals.

### 2.4. Qualitative Data Analysis

Three researchers (G.M., J.S. and A.M.W.) independently reviewed the questionnaire responses and developed themes based on their interpretations of the data. Data and identified themes were shared with Aboriginal community co-authors (J.F. and L.L.) and representative Aboriginal community governance bodies from each participating community to ensure themes identified during analysis accurately represented their experience. Emergent themes were discussed jointly by the research team to draft the manuscript. Questionnaire responses were re-reviewed frequently during the analysis to ensure the interpretation remained true to responses.

### 2.5. Ethics and Aboriginal Community Approval

Prior approval for this study was granted by Aboriginal community governance bodies (including local Aboriginal lands councils and community working parties) in each participating community. Ethical approval was granted by the University of Sydney Human Research Ethics Committee (project number: 2018/763).

## 3. Results

### 3.1. Respondents and Dog Ownership

Questionnaire responses were received from 85 dog owners participating in RSPCA ICCAHPs. Most of the respondents (71 of 85, 83.5%) identified as Aboriginal. No dog owners declined to participate. Nine communities that participated in an ICCAHP between September 2018 and November 2019 were included in the study (Table 1). Communities 5 and 9 had previously participated in similar RSPCA programs in 2014.

Most respondents had one or two dogs (Figure 1). Respondents frequently reported acquiring their dogs from friends and relatives “I got him from a litter off my mother’s dog. My brother’s dog, Fred, came from the same litter” (S49) or breeding pups themselves. Others had taken on responsibility for dogs from relatives and neighbours “Rex… He’s from next door but stays with us” (S41), “Because she’s my son’s dog and I couldn’t let her go. She’s family. I got attached to her.” (S52) Close to half of the respondents were return visitors to the ICCAHP from previous years (40 of 85, 47.1%). Most of these return visitors still had the dog from the previous year still living with them (33, 82.5%), including two dogs from communities that last received an RSPCA program in 2014. 

### 3.2. Relationships with Dogs

All 85 respondents said “yes” when asked “Do you think of your animals as part of your family?” Respondents most often kept dogs as companions (72, 84.7%), followed by as guard dogs (39, 45.9%), for kids (24, 28.2%), and because they needed a home (12, 14.1%). Dogs were considered an important part of family life “I have always had dogs. Homes need to have pets. Good company” (S24), “They are an integral part of this family” (S50), “We grew up with dogs. My family, my mum and dad always had dogs. My brothers would go up the river hunting [with dogs]” (S101).

Most respondents reported spending time interacting with their dog/s everyday (71 of 83, 85.5%). Indeed, dogs were a prominent feature in many respondents’ day-to-day lives “All day every day, they’re around us all day, they’re people dogs” (S96), “Doesn’t matter where you move in the yard, you’ve always got a dog with you” (S92). Most respondents’ dogs regularly accompanied them on excursions (e.g., on walks, hunting trips, drives in the car); 39 of 82 respondents (47.6%) took their dog/s out daily, 21 at least weekly (25.6%). Only 14 respondents said they took their dog/s out rarely (17%) and three responded never (3.7%). Several respondents described their dogs accompanying them on most if not all outings “They follow us, like, every time we go out of the yard. Every day.” (S100), “We go everywhere together, when we go to [the reserve] she always gets there first, she’s how they know we’re coming. We can’t surprise anyone! She follows the whole family and gets upset when we go different ways.” (S68), “He follows us all around town-the whole family” (S57). 

Respondents from Community 3 were significantly less likely to report interacting with their dog at least weekly: for example, by playing games or patting (ꭓ^2^ 47, df = 1, *p* < 0.01). However, respondents from this community were no less likely to go out with their dogs at least once a week (ꭓ2 0.34, df = 1, *p* = 0.56); when asked “how often does your dog go out with you? (e.g., on walks, hunting trips, holidays, drives in the car)”, five of six respondents said every day. None of the houses in this community had fences, and all respondents’ dogs were able to roam freely.

Respondents frequently agreed with statements about positive aspects of owning a dog (Figure 2). In particular, dogs were valued for their friendship “Greatest friend to have” (S60), “Almost constant companion” (S72), “They’re helpful. They reckon they’re man’s best friend and I believe that.” (S54) and for providing emotional support and connection “Someone to talk to when you got no one to talk to” (S57), “They help on lonely days when the missus is working” (S106), “They don’t judge me, and they love me unconditionally” (S21). Respondents’ dogs increasing personal safety was also highly valued with 53 of 85 respondents (62.4%) agreeing with the statement “my dog helps me feel safe”: “They let me know when someone’s around” (S58), “She makes me feel very safe” (S52), “Dogs are good for breaking up fights, stop the bad ones. Growled at police, wouldn’t let them in.” (S68) The majority of respondents agreed with the statement “my dog helps me when I feel stressed or sad” (52 of 85, 61.2%): “When I’m stressed, I go throw a ball with them” (S58), “When we’re sad we play with him. He likes to cuddle—he snuggles right in” (S59). Respondents’ comments often reflected their pride in caring for their dogs “It’s good to look after them” (S58), “I’m proud to look after my animals” (S57).

When asked to identify barriers to interacting with their dog/s, 62 of 82 respondents (75.6%) said there were none. Likewise, when asked if looking after their dog/s makes them stress or worry the majority of respondents (55 of 85, 64.7%) said no. Conversely, 26 respondents (30.6%) said they did worry about their dog/s sometimes and four respondents (4.7%) said they worry about their dog/s often. Reasons respondents gave for worrying about their dogs most often related to fear of injury and illness “When attacked by other dogs, worry about them being in pain or discomfort” (S23), “Them getting hurt, other dogs getting injured” (S81), “Fearful of others hurting them. Dogs getting injured” (S77), “Worry about them if they get sick.” (S79), concerns about dogs roaming “Running away, hit by car” (S74), “Worried about them getting into fights and getting out of the yard” (S90), “Worry about the cops who will pull out a gun and shoot her” (S68), “Kids opening the gate, letting the dogs out” (S101), “Out on the station leave baits around, frightened he’ll come back with some bait” (S79), and about the cost of adequately caring for their dog/s “The cost of food” (S21), “Food is expensive” (S26), “When I’ve gotta feed too many of them and I can’t afford to feed them” (S105).

### 3.3. Dogs in Relation to Housing

The largest proportion of respondents lived in community housing (43 of 82, 52.4%), followed by privately owned (21, 25.6%), other (8, 9.8%), NSW government housing (5, 6.1%) and private rental (5, 6.1%). The majority of respondents allow their pets inside their house (52 of 82, 63.4%); many allowed access to the whole house, while others dogs had their indoor access restricted (Figure 3). Others allow their dog/s inside sometimes “He comes inside when there are thunderstorms because he gets scared” (S57), “Dogs come inside when it’s cold or raining” (S66).

Almost half of the respondents’ dogs sleep inside (37 of 82, 45.1%); 15 respondents (18.3%) that answered the question “where does your dog sleep?” volunteered that their dog/s sleep in bed with their family “Spot sleeps in bed with me” (S105), “My little companion, she sleeps with me” (S47), “Our bed” (S72), “Three in the house in bed with us.” (S53). Respondents often reported that their dog can freely choose where they sleep “Anywhere they think is comfortable” (S50), “One on the floor at the foot of the bed, one whichever bed he chooses.” (S96) “Inside, wherever they choose” (S76). Other dogs, however, were doing their most important work overnight, warning their family of intruders “[he sleeps] at the front door. He chased someone out of the yard last night” (S108), “At night she lies on the verandah. She lets me know when anyone’s around” (S52).

Most respondents’ properties had fences that could effectively keep their dogs in and other dogs out (63 of 83 respondents, 75.9%), including all of the respondents from four of the participating communities. When asked how often gates are left open, 18 of 63 respondents (28.6%) with effective fences responded at least “sometimes”. In a separate but related question, “how often does your dog roam away from your property?” almost half (40 of 84, 47.6%) of the respondents said at least sometimes. The proportion of pets that roam varied between communities, with respondents from communities receiving the ICCAHP for the first time significantly more likely to report their dogs roaming at least sometimes (OR 4.75, 95% confidence interval 1.82–12.43) than respondents in communities that had participated previously. Several respondents’ comments suggested they value and actively facilitate their dog’s freedom and autonomy “She’s free to come and go. I tried keeping her on a chain once—it stresses her out.” (S46), “My dog is happy. He doesn’t have to be locked up or tied up, he’s free to do what he likes.” (S57), “In the morning we sit out there (on the verandah) and have a cup of tea then she goes walkabout up the road then she’ll come back. She goes out every day—to my daughter’s house, to my son’s house, then finds her way home.” (S52), “Phil goes out twice a day on his own, goes up the street to friends or relatives.” (S59) Other respondents described their dog roaming as the consequence of ineffective fencing “Bobby can get under the fence” (S57), “The property has high fences but Sally can jump them. She’ll go next door to play with Bruce” (S58). 

Respondents expressed a general reluctance to keep their dog/s on chains “I don’t believe in putting dogs on chains” (S21), “I tried keeping her on a chain once—it stresses her out, she got too hot” (S46). Only 22 of 81 respondents (27.1%) said they keep their dog/s on chains at least sometimes. The reasons given for using chains included: because fences were not secure (12 of 22, 54.5%) “On a chain to keep him from getting out” (S32), “Because I don’t have a fence” (S42), to prevent dogs from fighting with each other (5, 22.7%), to prevent dogs breeding (2, 9.1%), and to stop dogs attacking people (2, 9.1%).

### 3.4. Challenges to Dog Population Management

#### 3.4.1. Perceiving a “Problem”—A Case of Different Perspectives and Norms

When asked “Do you think there is a problem with dogs in your community?” 50 of 85 respondents (58.8%) said yes, 25 said no (29.4%), eight didn’t know (9.4%). Eight respondents (9.4%) from five of the nine communities reported being bitten by a dog in the previous 12 months “My mother got bitten by an unleashed dog” (S30), “Dogs attack during walks” (S84), “I got bit by a dog” [child, shows big scar on his leg] (S100), “Got bit breaking up a dog fight” (S79). Twelve respondents from five communities had been chased or frightened, “I got rushed by one maybe a month ago” (S74), “A bit frightened of dogs coming in yard. Frightened to go near them” (S52), “Baled up by 2-3 dogs” (S70). In total 16 of 85 respondents (18.8%) recalled an incident of being bitten or chased in their community in the last year, four of whom responded “no” when asked “Do you think there is a problem with dogs in your community”.

Some respondents commented that free-roaming dogs affected their ability to move around the community “[dogs] want to breed with Josey when she’s in season. The yard was full of dogs last week, some were really vicious. I just stayed in the house with the broom, couldn’t come outside.” (S52), “I would like to walk [my dogs] more often, worried about free roaming dogs in town” (S69), “Can’t walk the dogs in town—don’t want them getting in a fight.” (S58), “Couldn’t visit my old sister because of dogs not owned by her” (S60). Others commented on the danger to their own and other dogs “Dogs at [the reserve] attacked another dog one week ago” (S26), “They always rush each other” (S102), “Other dogs on the mission attack [my dog]” (S79).

There was no significant difference in the number of respondents reporting incidents of being frightened or bitten by dogs between communities that had received an ICCAHP in the previous four years and those participating for the first time (OR 0.85, 95% CI 0.23–3.10 and OR 3.27, 95% CI 0.72–14.76 respectively). However, respondents from communities that had previously received an ICCAHP were significantly more likely to think there was a problem with dogs in their community (OR 3.17, 95% confidence interval 1.20–8.32). Comments like the following from a respondent in Community 2 (who received their first ICCAHP during the study period) after saying “no” when asked “do you think there is a problem with dogs in your community” and “yes” to both having been bitten by a dog in the last 12 months and having been chased or frightened by a dog in the last 12 months suggest normalisation of poorly managed dog populations in communities without accessible veterinary services might contribute to this cognitive dissonance: “It’s been like this most of our lives, we all grew up on the mission with animals. I think we’ve all been bit at some stage, but that never stops us going back for a dog” (S99).

#### 3.4.2. Stray and Extra Animals Are Part of Life

More than half the respondents (46 of 85, 54.1%) said “yes” when asked “do you care for any extra animals that you don’t own?” This included people caring for animals belonging to neighbours and relatives and people feeding and caring for stray (i.e., free roaming) animals. One respondent from Community 2 summed up the overall feeling of most respondents in their response: “It’s all one family, everybody looks after everybody else” (S96). Several respondents care for animals of neighbours or relatives on a temporary or semi-permanent basis “Brother’s dog and auntie’s dog lives with us, he won’t go home.” (S49), “I look after my sons’ dogs when they go away or go to work” (S96), “Just permanent boarders. Janice used to walk back and forwards, between here and my sister’s house but now stays” (S101). Mostly, stray dogs are welcomed and fed: “Two small dogs, one male, one female hang around my place, I feed them” (S87), “Other dogs just come over here, if we’re eating tea, we’ll give it to them.” (S52), “Homeless dog come around when feeding time” (S42), “Stray dogs. If they come in give them some biscuits” (S95), “Strays, if they seem a bit hungry, I feed them.” (S103) Some respondents’ comments highlight the overlap between dogs that are genuine strays (unowned) and dogs that are owned and roaming “Neighbour’s dog gets food and welcomed” (S21), “Family dogs come over, share feed” (S97), “Pauly [a neighbour’s dog] comes around and has a feed” (S57), “Sometimes next door’s dogs [come over], three to five small dogs” (S77).

#### 3.4.3. Barriers to Accessing Veterinary Services Go Beyond the Distance and Cost

When asked “when do you take your dog/s to the vet?” over one third of respondents (32 of 85, 37.6%) said “none of the above”; of the remaining, 25 said they would take their pet to the vet if they were injured “If you gotta do it you gotta do it” (S106), “Whenever something is wrong with them” (S81), 23 if they were sick “Will go if she’s sick” (S99), “When they were sick once 10 years ago. It cost a lot of money” (S106), 23 for vaccinations “Took a dog for a parvo needle once” (S95), 15 for de-sexing, 15 for check-ups (respondents could choose more than one response).

When asked “do any of these things stop you taking your pet to the vet?” most respondents (60 of 85, 70.6%) identified at least one barrier to accessing veterinary services, the most common being cost (47 of 85, 55.3%—“the vet can be expensive” (S81), transport (28 of 85, 32.9%) and time (15 of 85, 17.6%—“time can be a barrier if you’re working. Having to take time off work to travel 3–4 h to the nearest vet” (S47)). Five respondents (5.9%) also reported that their animal was difficult to get to the vet; “too big to get in the car” (S102) “Bingo is too big to get in the car! It’s also hard to get him in because he’s scared” (S54). Several respondents reported nursing dogs back to health after injuries or illness “Sometimes I just do it myself” (S105), “She has been hit by a car and beat-up by roos, she had bad wounds both times, we baby’d her back to health” (S68). Nineteen responded “none of the above”; however, five of these respondents had never taken an animal to the vet with one commenting “don’t need it” (S77) and another “don’t know what the cost would be like—never been” (S54).

#### 3.4.4. Dogs Breed Frequently and Often Unintentionally

Almost one third (27 of 85, 31.8%) of respondents reported their pets having a litter in the last 12 months. When asked if the litter was planned only three of the 27 respondents (11.1%) said “yes”, however, most (17 of 24, 70.8%) said “No, but it was a happy surprise”, rather than an “unwelcome surprise” (7 of 24, 29.2%) “My dog had a litter of puppies recently, I’m happy now but wasn’t really keen.” (S107) “I didn’t know she was having them.” (S95) Respondents comments reflect the significant fertility and fecundity of individual dogs “Other dog I give to my grandma had a litter, she’s about 13–14 now, she’s had about 10 litters” (S103), “Buddy and Bella recently had a litter. Buddy has fathered a lot of puppies.” (S61), “Ellie has had two litters of twelve puppies each” (S68). Finding homes for litters of puppies can be a problem “Bluey had pups, gave them all to the dog catcher except Max” (S56). Respondents from communities that received an ICCAHP for the first time were significantly more likely to have had an unplanned litter in the previous 12 months than respondents from communities receiving an ICCAHP for the second or subsequent year (OR 3.54, 95% confidence interval 1.32–9.50).

#### 3.4.5. Dog Population Turnover Is High

Several respondents described having dogs die from potentially preventable deaths. Many of these were puppies “Joey had a litter of puppies over the summer but all died soon after they were born in the heatwave. It was very upsetting for me and for Joey.” (S52), “Frank may have fathered one litter last year, but all the puppies died” (S59). Other respondents described having dogs die of accidents and injuries including dog attacks, snake bites “lost one pup to a snake bite” (S53), injuries from wild pigs “I had one dog for seven years. She died earlier this year. I found her in the river, she had been gored by pigs, she was trying to get back home” (S103), baiting and road traffic accidents “We had a Maltese, he was my daughter’s dog. She carried him everywhere like a baby. He slept in my daughter’s bed, dressed in a nightie. He was hit by a car and killed. Everyone was very upset.” (S68).

Parvoviral enteritis (parvo) is a highly contagious, often fatal but vaccine-preventable disease causing severe haemorrhagic gastroenteritis in dogs, especially puppies [24]. Self-reported parvo was a major cause of death amongst dogs in several communities, however, this likely represents overlap between true cases of parvoviral enteritis and illness due to other diarrhoeal diseases. The question “have you ever had a dog get sick or die from parvo?” was added after the first 21 questionnaires were received so was unanswered in Community 3 and the first year in Community 4 and Community 7. This question was added in response to several early questionnaire respondents mentioning parvo. Overall, 33 of 64 respondents (51.6%) had had a first-hand experience of a dog with parvo. Respondent experiences varied between communities. Communities 2, 4 and 7 have current and ongoing disease, which has resulted in the loss of many dogs and puppies “About three months ago. Four died, one survived, they were about nine months old” (S88), “Too many, last one early this year. We have lost three dogs in the last year.” (S90), “One puppy sick now, another puppy (littermate) died last week” (S92). In contrast, respondents in Communities 6 & 8 had not seen cases of parvo for several years “Years ago, pup died from parvo—probably ten years ago” (S53), “15 years ago—puppy who was about six months old” (S58) “Parvo used to be bad once, been quiet lately.” (S54) 

Several respondents described repeatedly losing dogs to parvo over years: “Over the years we have lost too many to remember.” (S90), “More than ten dogs, some died, some didn’t” (S74), “Have had many puppies die from parvo” (S91), “About three or four, when one died got another and then it died.” (S68) Often whole litters die “Previous litter all died, 12 puppies.” (S95), “Nine out of ten puppies of the last litter died of parvo at around 2-3 months old” (S107). Respondents mentioned the emotional toll of losing puppies to parvo: “Young pup 2–3 months old, was sick for 4–5 days then died. Very sad, stressful time.” (S93) While mortality from parvo is high, some respondents reported successfully nursing their dogs through the illness “The dogs I look after now had parvo when they were younger. I didn’t think they would survive but brought them through with sterile water to keep them hydrated” (S81), “My pups before, three out of four died. They were 5–6 months old. Oldest survived, just kept feeding him coke” (S103), “Last litter of puppies recently all but two out of nine puppies died of parvo. Had a lot of dogs with parvo, lost two other dogs, mostly nursed them through” (S105).

## 4. Discussion

This study is the first to investigate relationships with dogs in diverse Aboriginal communities across NSW and the first to comprehensively investigate underlying causes of dog population management challenges and barriers to accessing veterinary services in Aboriginal communities. The findings demonstrate that dogs have an important role in these communities. Many Aboriginal people love and value their dogs, especially for companionship and safety. However, major barriers exist to accessing basic veterinary services in many Aboriginal communities making dog population management a particular challenge. Our findings emphasise the need for dog health interventions in this context to adopt a different approach to mainstream initiatives and explain why such interventions need to be tailored for individual communities.

Respondents in this study primarily value their dogs for companionship and for protection of person and property, consistent with findings from earlier studies in remote Aboriginal communities in Australia, most of which have been conducted in isolated communities in the Northern Territory or Queensland [11,12,13]. The potential benefits of companionship with dogs have been well described. Dogs can provide positive psychological, social and physical benefits for their owners, and can be a valuable support, particularly at times of crisis and loss [2,3]. Dogs can provide a sense of family, a sense of responsibility and purpose as well as being a catalyst for social interaction and connections [4,25,26]. Interestingly, despite significant challenges to dog health and welfare, and the strong bonds between respondents and their dogs identified in this study, relatively few respondents reported worrying about their pets, and very few could identify anything (e.g., being unhealthy, having parasites, being dirty) that would stop them interacting with their dog. This is an important contrast with previous studies in mainstream populations that have consistently noted negative aspects of dog ownership, particularly worrying about pets, grief at loss of pets and frustration and anger at misbehaviour [2,25], suggesting that dogs have a net positive effect on health and wellbeing of Aboriginal people and that this effect might be greater for Aboriginal people than for other populations. The reasons for this apparent contradiction might be related to different perspectives on the role of dogs in the community, which is further discussed below.

The dogs of study respondents are genuinely considered part of the family. All respondents in this study reported they consider their dog is part of their family compared to only 62% and 63% of dog owners asked the same question in an Australia-wide survey and in the United States respectively [27,28]. The dogs in the communities we studied often spend much of the day with their families and many have unrestricted access indoors, even sharing beds with the people in their homes. This has important implications for environmental health. Dogs are a potential source of zoonotic diseases and parasites including brucellosis, leptospirosis, Q fever, *Microsporum canis*, *Ancylostoma caninum* (hookworm), *Toxocara canis* (roundworm), Giardia, *Sarcoptes scabiei* [29,30,31]. The role of dogs in transmission of disease and parasites directly to humans is likely to be less important than other factors such as direct transmission between people and over-crowding [32,33,34,35]. However, dogs can exacerbate existing environmental health challenges, for example by faecal contamination of yards, spreading contamination resulting from faulty plumbing and sanitation systems, bringing dust and dirt inside homes, spreading garbage from uncovered bins and contributing to minor trauma in their human family through scratches, bites and parasite transmission (fleas, ticks, *Sarcoptes* spp. mites). Environmental health challenges related to dogs are likely to be amplified by larger numbers of dogs, especially if dogs have poor skin health, are hungry and are not well behaved. Hence, dog health and population management should be considered an integral component of household and community environmental health initiatives and not just the regulatory responsibility of local government. Dog population management is often expressed as an environmental health priority by communities, presenting an opportunity to use dog health programs to build relationships and trust between communities and environmental health service providers.

The cost of basic veterinary care, access to transport and distance from services were identified in this study as the major barriers to accessing veterinary services. Consequences resulting from a dog health service gap have previously been highlighted in remote Aboriginal communities in Australia [17,18]. This situation is mirrored in remote First Nations communities in Canada [14,36]: both feature a history of colonisation with ongoing consequences for Indigenous populations; in both instances, First Nations communities can be remote and isolated and experience challenging natural environments [11,37]; both often rely on access to veterinary services from outside their communities and both have historical relationships with dogs that differ from the mainstream [12,13,37]. The result is significant preventable dog suffering and death from disease as well as compromised community health and safety due to dog over-population. The present study highlights that lack of accessibility of veterinary services in Aboriginal communities in Australia exists regardless of the distance from a veterinary practice. Comparable challenges to dog health and welfare and identical barriers to accessing veterinary services were encountered across communities classified as very remote, remote, outer-regional and inner-regional, regardless of whether a veterinary practice was located close to the community. This has also been observed in Canada [36] and highlights the importance of social and cultural barriers to accessing services. A parallel scenario exists in Australia with Aboriginal people under-utilising mainstream health services. This has been attributed to mainstream medical services delivering rigid and standardised models of care, which aligns with the dominant cultural group and passively or actively excludes individuals with other cultural identities [38], a situation that might also be applicable in a veterinary setting. 

Our findings reinforce the ideas discussed in previous studies [11,12] that Aboriginal people consider their dog’s freedom and autonomy to be very important. Respondents in this study want their dogs to have the same quality of life that they do, they want them to be free and consider this part of their rights. This extends to concepts of “ownership”—where ownership does not necessarily imply control—a concept very different to the often paternalistic or possessive mainstream understanding and one that is often at odds with requirements for “responsible pet ownership” in the relevant legislation [11]. This difference of perspective likely explains a number of the findings of the current study, including findings that can appear contradictory: respondents being reluctant to prevent their dogs from roaming freely whether by confining them by fences or chains; respondents not reporting worrying about their dogs, despite caring deeply for them; respondents not necessarily seeking veterinary attention in the face of dog suffering, despite being distressed by dogs with injuries or illness. A general opposition to controlling dogs, as articulated by Musharbash [12], is likely also an important reason that permanent procedures such as surgical sterilisation are often declined, and why temporary and reversible alternatives such as chemical contraception might be a more palatable means of population control.

Our findings also suggest that low standards of dog health and welfare have been normalised in some communities, consistent with findings of previous studies on dogs in Aboriginal communities in Australia [11,12]. This extends to a lack of engagement with veterinary services. As stated by Musharbash [12], as regards dogs in the remote central Australian Aboriginal community Yuendumu: injured dogs are cared for as best as people can manage but are often left to their own devices. This might explain why several of the communities in the present study do not consider dog population management in their community to be a problem, despite high incidence of preventable disease and suffering and high incidence of dog attacks on people and animals. The statement by one study respondent from Community 2: “It’s been like this most of our lives, we all grew up on the mission with animals. I think we’ve all been bit at some stage, but that never stops us going back for a dog” (S93) represents this concept. Normalisation might explain why respondents in communities that have been receiving dog health interventions over several years are more likely to identify problems with dogs in their community than respondents from communities visited for the first time. This suggests consistent service provision in these communities is gradually shifting norms and changing expectations around dog health and dog population management. These findings emphasise the importance of empowering and resourcing communities and individuals to recognise dog management issues and facilitating the development of community-led plans to address them [37].

Our findings demonstrate a subsection of the community will not seek out veterinary attention for their dogs, even if it is present locally and free. This should be considered in relation to the wider social context. Some Aboriginal people in western NSW, particularly in the communities prioritised for ICCAHP services, experience very poor socioeconomic conditions [39]. Dog health and welfare, unsurprisingly and justifiably, is often overlooked as a result of competing priorities. Hence, sustaining meaningful improvements in dog health and welfare standards and in dog population management will depend on successful engagement and knowledge exchange with this cohort of dog owners and the removal of barriers to accessing services. This has previously been acknowledged as a vital component for the success of dog population management interventions [20].

Another important consideration for dog health interventions highlighted in this study is that, despite some common themes, every participating community is unique in its specific dog health and population management challenges and in its perspectives on the role of dogs in the community. This is particularly pertinent in Australian Aboriginal communities, which have considerable diversity and differences in cultural identity between geographic locations. Consequently, rather than adopting a one-size fits all approach, dog health and dog population management interventions are likely to be more effective when they directly address the specific needs of each individual community. This reinforces the need for effective engagement with the communities in planning companion animal health interventions to identify specific challenges, priorities and desired outcomes.

Despite not intentionally aiming to evaluate the ICCAHP, this study demonstrates some preliminary evidence of benefits of this dog health intervention. A reduction in unplanned breeding and fewer dogs being allowed to roam in communities visited over several years compared with communities visited for the first time suggests the ICCAHP is delivering benefits for dog population management. The high proportion of participants that returned to the ICCAHP in subsequent years also suggests the program is effectively engaging these communities and assisting people to keep healthy pets for longer.

This study has some limitations. Convenience sampling employed to recruit participants may have introduced some bias to the study findings, while a perception that the study was linked to the ICCAHP might have influenced the way some questions were answered. In addition, only community members that own dogs were interviewed for this study; hence, views on dog problems in the community will likely be biased. A very different perspective on dogs and their role in communities would be expected from non-dog owners.

## 5. Conclusions

This study demonstrates the important and positive role that dogs play in Aboriginal communities. However, a major veterinary service gap in these communities, whether or not they are located remotely, presents an important challenge to maintaining healthy and stable dog populations. An effective response requires multiple stakeholders, including public and environmental health practitioners, housing managers, local government and veterinary service providers to work collaboratively with community partners. Different perspectives on the role of dogs and relationships with dogs necessitate a different, and culturally inclusive, approach to dog management interventions in order to achieve long term gains. These findings demonstrate that dog health interventions in this context must reach out directly to individuals in target communities, they must be consistent and sustained and must prioritise building trust and interpersonal relationships. Community engagement and effective knowledge exchange are important and must emphasise empowering communities to make their own decisions around the management of their dogs rather than attempting to impose mainstream dog ownership and management values.

## Figures and Tables

**Figure 1 animals-10-00810-f001:**
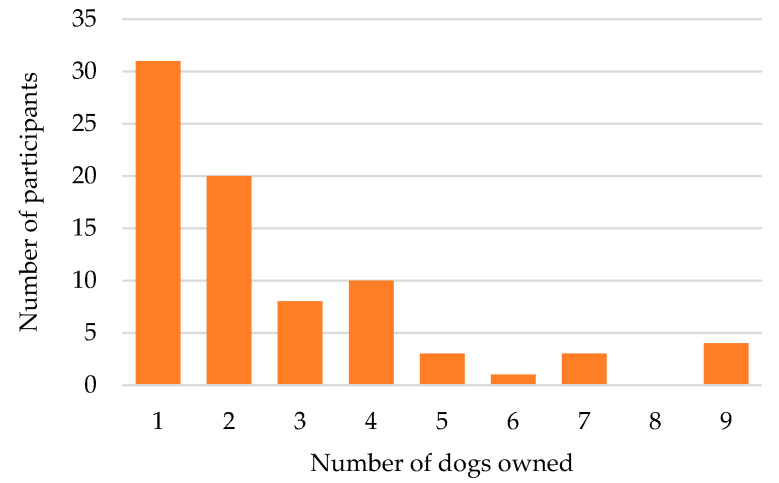
Number of dogs owned by questionnaire respondents in nine Aboriginal communities in New South Wales, Australia.

**Figure 2 animals-10-00810-f002:**
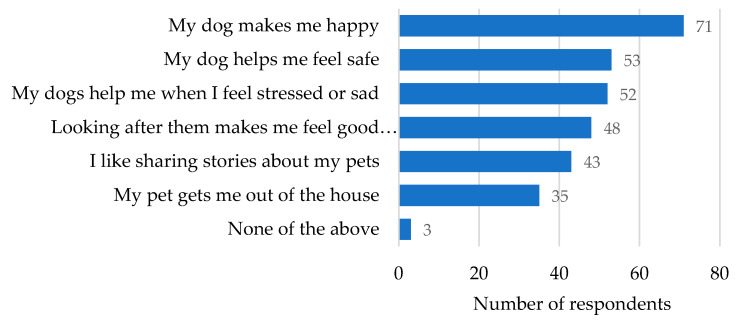
Positive aspects of dog ownership identified by dog owners in nine Aboriginal communities in New South Wales, Australia.

**Figure 3 animals-10-00810-f003:**
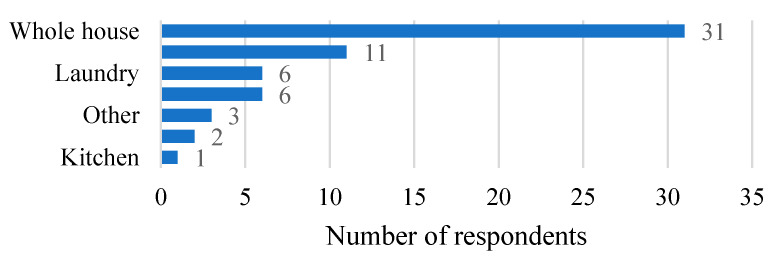
Access of dogs indoors reported by respondents to a questionnaire about relationships with dogs in nine Aboriginal communities in New South Wales, Australia.

**Table 1 animals-10-00810-t001:** Dog owners interviewed from each of nine Australian Aboriginal communities and the year communities received the dog health intervention.

Community Number	Number of Respondents	ICCAHP ^1^ Years			
		2016	2017	2018	2019
1	5		×	×	×
2	14				×
3	6			×	
4	11		×	×	×
5	5				×
6	2	×		×	×
7	13	×	×	×	×
8	23			×	×
9	6				×

^1^ Indigenous Community Companion Animal Health Program.

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
