# Peer review of "“They Reckon They’re Man’s Best Friend and I Believe That.” Understanding Relationships with Dogs in Australian Aboriginal Communities to Inform Effective Dog Population Management"

_animals, 2020, doi:10.3390/ani10050810_

Round 1

Reviewer 1 Report

This manuscript is well written and researches an important aspect of dog population management in Australian remote communities.

Major Issue (Methods):

The questionnaire-based research surveyed only the dog-owning participants in 9 communities that were receiving subsidised (free?) veterinary care. The authors identified the issue that only dog owners were surveyed as a limitation of the study (noted at the end of the manuscript), but do not mention bias in relation to participants being recipients of the subsidised veterinary care. What was the relationship between the researchers and the care providers? If the researchers undertook the interviews during (or in association with) the health care program, this is likely to have created bias in that the participants may not wish to jeopardise their veterinary care by giving negative feedback. How did the researchers manage this inherent bias? In the eyes of the participants, were the interviewers perceived as independent, or as part of the dog health program? 

Minor Issues:

Title:

The title is too long and needs to more accurately reflect the research. "Relationships with dogs in Australian Aboriginal communities – understanding different perspectives" - this part is fine. The effectiveness of dog population management was not evaluated, so should not be in the title. The quote should also not be in the title.

Introduction:

L 49 - change 'our' to 'human' (or similar)

L 50 - change 'is' associated to 'has been' or 'can be' associated (soften wording).

L 50 - change 'maintaining' dog population to 'managing' dog population - presumably you aren't suggesting to maintain status quo when there is overpopulation and poor health, for instance. 

L 84-85 - Dog population management (insert: 'ideally') entails humane control of dog population size and growth, (insert: 'including') management of free-roaming dogs.

Materials and Methods 

2.1 Study Context - provide a little more information and detail about what services the ICCAHPs provided, and whether the services were free or subsidised? 

L 120 - More information needed about how the participants were recruited. Include limitations here, and what measures were taken to address potential biases. 

Results

3.1 Respondents (insert:) 'and dog ownership'

L 153-157 Too much information that is also presented in the table (Table 1). Suggest deleting all of these lines, and direct reader to the table for details. 

L 164-173 The first sentence is confusing, and the relevance of this whole paragraph is not clear. Suggest removing the entire paragraph - or rewording with greater clarity and purpose.

L 344 - The relevance of this whole paragraph is questionable, as it is about cats, not dogs. Suggest removing references to cats, here and elsewhere, as this is a distraction from the aims of your research.

L 387 - Change wording of heading. Suggest"Disease and Mortality" or similar.

L 391 - remove reference to kittens

Table 3 - This table is not necessary. There is sufficient information in the text. 

L 424 - mortality from (insert: 'presumed') parvovirus 

Discussion

The discussion is well-written, flows well, and makes some very good points - but is rather long. Consider the possibility that you may loose interest in your readers with a long discussion, and whether there are places that this can be made more concise without losing the important discussion points and the flow of ideas. May not be possible - but worth considering?

Conclusion

Rather than re-iterating earlier statements, this is an opportunity to break new ground - make recommendations, suggest future research. Perhaps make suggestions for future management based on your findings?

General

The stated aims were to (1) gain a better understanding of dog ownership and human-dog relationships in Aboriginal communities and (2) identify barriers to accessing veterinary services in these communities to "inform the design and delivery of best practice dog population management interventions". 

I believe that attitudes to dog ownership were also evaluated, and these revealed some similarities and differences compared with mainstream views in Australia. 

The manuscript would be strengthened by providing suggestions or recommendations as to how the "design and delivery" of current dog management programs could be improved based on the unique characteristics of dog ownership/relationships (and attitudes?) that were revealed by your findings. The conclusion could be rewritten to achieve this.

Reviewer 2 Report

Thank you for the opportunity to review this paper. The issue of animal and human health is important, and given the well known gap in living standards and life expectancy between indigenous and non-indigenous people in Australia, there is great value to be gained in improving the health and wellbeing of companion animals such as dogs. However I would encourage the researchers to think carefully about the power dynamics that are inherent in any research that explicitly seeks to problematise indigenous communities as it is hard to avoid replicating and reinforcing colonial relationships, even with all of the best intentions in the world.   Overall I think this research has value for the journal’s audience and I recommend resubmission once the following points are addressed.   Title: not sure that the title accurately reflects the conclusions of the research - suggest a slight amendment  "understanding different perspectives MAY BE the key to effective dog population management"   Line 65 - delete unnecessary comma Also here - what is meant by ‘readily replaced’? Biologically or figuratively? The issue of dingoes is one of some controversy so it pays to be clear here. Line 91 - in text citation inconsistent 99 - not just remote communities - include the range as described in ABS data later quoted   Part 2.2 - include some information here about how the questionnaire itself was developed - particularly given the claim to participatory action research. Was the questionnaire piloted first? If not, why not? Line 127 - not clear whether comments were recorded verbatim with a voice recorder or written down? If audio recorded, were they then transcribed?    Results - Section 3.1 is confusing to read - the table is essential - try to rewrite for clarity  Same again for sentence line 164-166 - confusing 166 - Change in terminology from ‘dog’ to ‘pet’ - why? 169 - When quoting directly from respondent, include a code or identifier so results can be distinguished - otherwise it is hard to know whether the quotes adequately reflect views collected across the sample   175 - “only” carries some subjectivity - suggest you stick to reporting the data without interpretation here 175 - replace semi colon with colon after Figure 1   Section 3.2-  be careful of slipping tense between past and present in reporting results - as this research has already occurred, stick to past tense throughout. This is a problem that reappears throughout the results.   Line 209-210 - appears that some explanation for the results for community 3 are being included here, but I wonder if these features (no fences etc) are different to those in the other communities? If not, consider whether these points are relevant to include here or whether you might explore these possibilities in the Discussion.   Line 243 - why capitalise ‘community’ housing? Use the correct organisational name or keep as lower case.   Overall - The reporting of results is too cluttered. Given this is qualitative research, it is not necessary to include a descriptor (eg almost half) as well as the number of responses and a percentage. Suggest that you chose one metric and report this consistently to streamline the results and increase comprehension. You can more effectively show the percentages in a table or graph.   Section 3.4.1 - Along with this - unless you are going to provide a more detail description of the differences and commonalities between the communities, I don’t think it adds to the reporting of results to go into great detail about how many people from which community said what - the interesting part is why these differences might exist. As currently reported, it is just a roll call of numbers and I think the story is better told with overall totals as detailed in line 304.   I find it hard to identify the cognitive dissonance that is mentioned at line 323 - it seems that the author is suggesting that participation in ICCAHP leads to a new norm of accepting problem dog behaviour - is this correct? If so, this is very important and needs to be emphasised in the Discussion. If not, suggest you carefully rewrite this section as the results are not clear.   Line 398 - include the abbreviation ‘parvo’ directly after the scientific name rather than introducing in the next sentence   Line 472 - do not capitalise ‘local government’ unless referring to a specific organisation   Discussion section: The discussion talks about the environmental health risks that might be exacerbated by dogs but doesn’t really engaged with the structural disadvantage experienced in Aboriginal communities that lead to over crowding, poor sanitation, etc.    It would be interesting to understand more about the comparative insights gained from the study, such as the detail at line 455 about corresponding values in the general Australian and US population. Otherwise the findings lose an important reference point - for example, how do the barriers identified correspond to those experienced in other human populations that have low SEC and are distant from veterinary services?  An important analytic question to keep in mind throughout the Discussion is whether the findings are unique to Aboriginal communities? Or are they likely to be replicated in non-Aboriginal communities living under similar conditions?    The recognition of social and cultural barriers to accessing health and/or veterinary services is important (line 487) however given the ICCAHP program provided the context for this study, I would like to see some detail about how (or whether) these barriers are addressed and whether the program has any evidence to support the need for a more culturally appropriate dog health program.    Line 550 - suggest replacing ‘challenges’ with ’needs’ as this is less leading    Final two sentences need to be examined for punctuation as currently the last sentence hangs on its own.    General points: Include a map for international readers  - you don’t need to identify the communities explicitly but at least show the region and how distant it is from capital cities Include a bit more descriptive detail about the communities up front Simplify the reporting of results and consider using more tables or graphs to avoid cluttering the text.

Reviewer 3 Report

This study has looked once more at the behavior of Aboriginal populations as they relate to the dogs in their communities. An important finding is the strongly held belief in Aboriginal communities that dogs have their own rights to be free and to be allowed to decide their own quality of life and relationships within these communities, rather than be controlled by the dog health and population management imposed or recommended by ourselves.  

The paper suffers, however, from potential duplication of prior work on this topic and poorly presented and  incomplete references.

Specific suggestions:  

Title: suggest removing the Quotation

Line 67. What is a "skin name" ? Please explain. Line 89. Wording can be streamlined -- too many "and"s.

Discussion: Please explain in what way the current study differs and is novel from what has been performed before with Aboriginal communities. The references cited as relevant in Lines 447-452 do not appear to be referring to Aboriginals. Lines 477-478. This statement is interesting Can you elaborate about the health service gap of Aboriginal communities in Australia mirroring the Canadian First Nations ? 

References : Need major reworking. Clarify Refs 2 and 23. Is there a book or monograph title for Ref 23 ? 

Refs 4, 5, 27, 31, and 34  -- Journal title? 

Refs 5, 7, 27, 34. Contain incomplete or extra words or letters.

 Refs 6, 20. Please spell out the abbreviations for Gfk and ICAM .

Ref 7. What is B6.1 ?  When was it accessed ? 

Ref 14. Title of Journal -- first letter should be caps. No need to cite the French sub-title.

Refs. 16-18. Journal Tit'e spelled out (but missing first letter caps) in Ref 18 but abbreviated in Refs 16 & 17. Please be consistent. 

Ref 39. It author name correct ?

Reviewer 4 Report

It is a well done research on owner/community and dog relationship very different from that of Western culture.  The perspectives and attitudes of Aboriginal people are described well and also at the same time, pointing out some of the animal welfare issues that could overcome with the help of veterinary medicine.

A question about the participants.  How did you treat those who did not identified themselves as Aboriginal in the survey.  It seems that authors included them in the study.  For readers who are not familiar with Aboriginal population in Australia may be confused about this. 

There are some suggestions I would like to make  on how the results can be presented.  It is up to the authors to change it or not.  Fig. 1 is an important data, but it is already described in line 172 to 180.  Instead of the number of dogs they keep, results presented in line 232 to 240, indicating what they worry about.  These reasons are different from those living in areas with dogs kept under control and with frequent veterinary visits.  

Comments about Community 3 being notable exception may not be statistically correct.  Do you have any statistical evidence to support this?  Since the number of respondents in communities vary, it may not be very appropriate to compare between communities. If this community is different in terms of not using fences to keep their dogs, the comments about this should be included in the section starting 267.

There are some judgmental phrases which can be replaced with more descriptive ones.  When talking about fencing in line 264, the authors use the phrase 'the fencing quality is generally good'  This implies that authors are using their sense of what is good or bad rather than accepting their norms of how they want to keep the dogs.  Authors also use phrases 'without adequate fencing' in line 267.  It could be described as complete or firm or something like that to describe the fences.  

Another phrase that sound judgmental is the subtitle, 'Life is cheap' at line 387.  Phrases like 'high mortality rate for puppies'  or 'short life expectancy' could make is more scientific and objective. 

Since authors refer to the different perspectives and norms Aboriginal people have, and pay respect to their attitudes towards dogs, describing the way they treat dogs and care for dogs should be objective and not judgmental.

Round 2

Reviewer 1 Report

The authors have appropriately addressed the concerns raised by this reviewer.

Reviewer 3 Report

This revised manuscript has satisfied the concerns of this Reviewer. Thank you !